# Using a Unique Retaining Method for Building Foundation Excavation: A Case Study on Sustainable Construction Methods and Circular Economy

**Tai-Yi Liu** [1],*[iD]**, Shiau-Jing Ho** [2]**, Hui-Ping Tserng** [2] **and Hong-Kee Tzou** [1]

[1]  New Asia Construction and Development Corporation, 15F, No. 760, Sec. 4, Pade Rd., Taipei 10567, Taiwan;
tzou@newasia.com.tw
[2]  Department of Civil Engineering, National Taiwan University, No. 1, Sec. 4, Roosevelt Rd., Da-An District,
Taipei 10617, Taiwan; jessieho821@gmail.com (S.-J.H.); hptserng@ntu.edu.tw (H.-P.T.)
*  Correspondence: tylaytpe@ms9.hinet.net; Tel.: +886-88-691-000-9129

**Abstract:** The selection of a retaining method during the excavation of building foundations is always of paramount concern to engineers. In general, the application and use of steel H-shapes are typically practiced by designers to form the entire retaining system; however, sustainability issues, including carbon emission reduction, environment protection, material consumption, and resource circulation, are being increasingly considered when developing a new project. The Linkou Public Housing Project (LPHP), located in New Taipei City, Taiwan, is introduced in this paper to present a sustainable soil-retaining method that also exhibits the principles of a circular economy. The triangular shape of the foundation zone of the LPHP led to difficulty in setting the horizontal strut H-beam system. In this project, the "Anchor Pile with Steel Cable System (APSCS)" was adopted to retain the 11.5 m depth excavation for the LPHP foundation construction. The prime contents of the soil in the Linkou district comprises a laterite–gravel layer mixed with brown silty and sandy clay, with a groundwater level (G.L.) of −25 m. By adopting the sustainable APSCS method, the excavation of the LPHP foundation was safely completed. Approximately NT $350 million in direct and indirect costs of construction was saved, and the duration of the work was reduced by up to 90 days. Furthermore, the carbon emissions were reduced by 677.6 tons due to the diminished use of the steel H-shaped materials. The authors concluded that the use of the APSCS method in the LPHP was successful and it was a valuable reference for other similar projects. Moreover, the authors presented another retaining-system failure case, which was located near the LPHP site, to compare the success of the LPHP.

**Keywords:** case study; circular economy; LPHP; foundation excavation; APSCS; laterite gravel; retaining-system failure

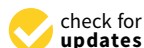

## 1. Introduction

A couple of retaining methods can be selected for a building's foundation or for other excavation works on concrete structures [1–7]. The most popular methods are diaphragm walls, steel sheet piles, pre-stressed anchors, PC piles, and steel H-shapes/rails combining the wooden boards. Except for pre-stressed anchors, all of these methods need to be performed using horizontal steel H-shapes as the strut members in order to form a functional retaining system. Figure 1 shows some representative photos for the above-mentioned soil-retaining methods.

To present a clearer picture for this paper, the authors illustrated the research framework to show the logical development of this study, as shown in the Figure 2.

The authors were aware that each type of retaining method usually required a large amount of money and would take a long time to construct. A couple of major concerns, such as risk management, impact to the environment, construction duration, construction

costs, use of materials, and energy consumption, always influence the selection of the construction method. Thus, approaches to improve safety concerns and risk reduction, environmental impact, optimization of construction procedures, and minimization of material consumption, are seriously discussed among engineers.

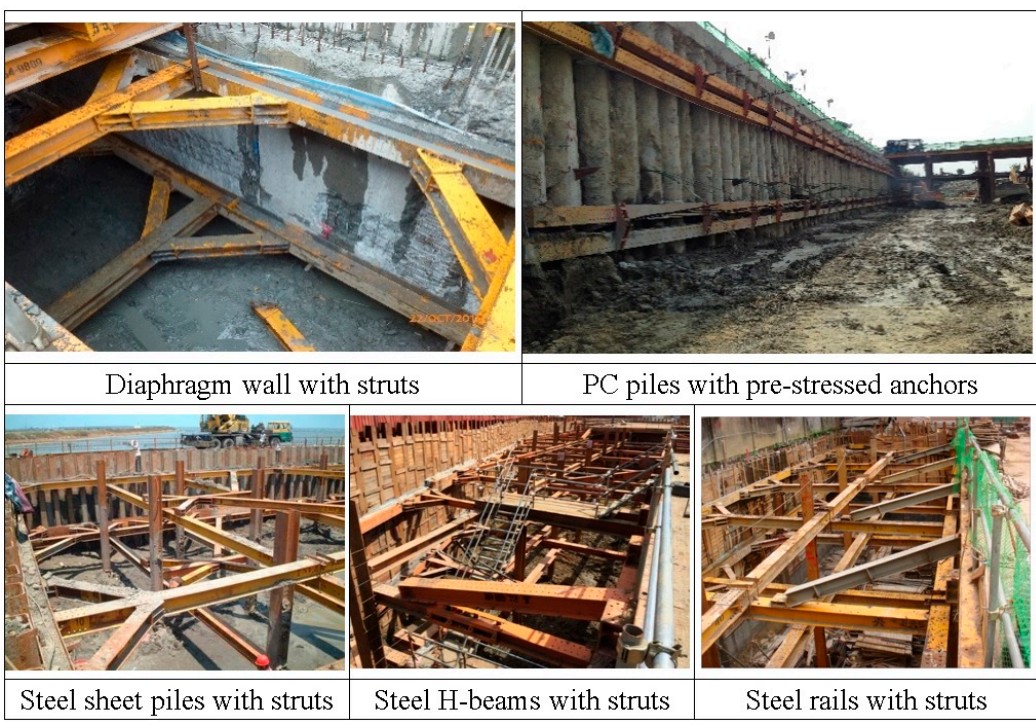

Diaphragm wall with struts      PC piles with pre-stressed anchors

Steel sheet piles with struts    Steel H-beams with struts    Steel rails with struts

**Figure 1.** Representative photos for foundation excavation retaining systems.

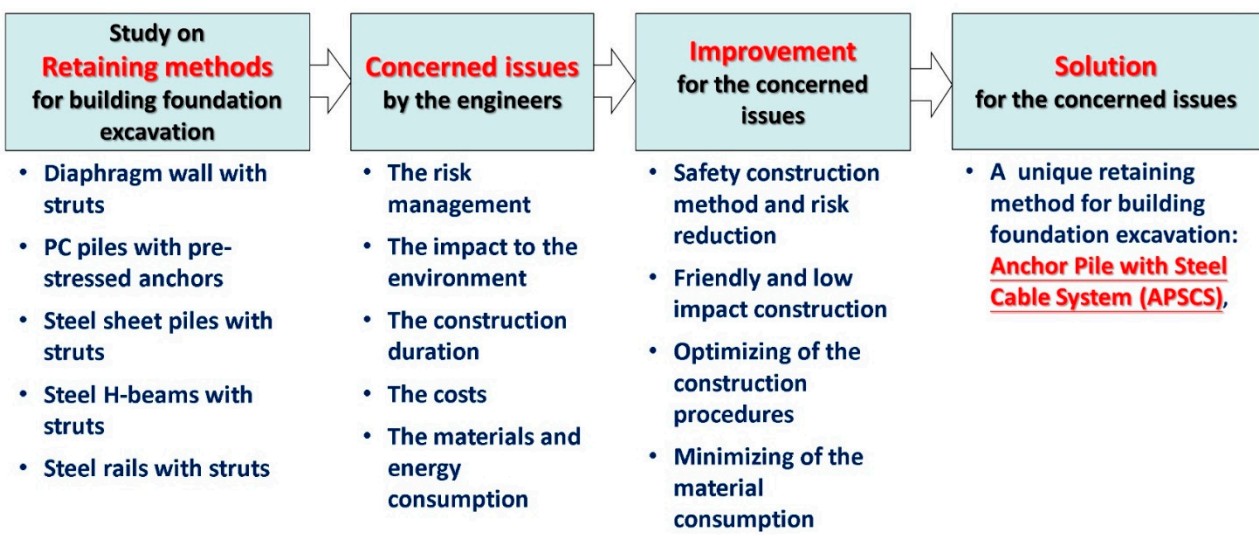

**Figure 2.** The research framework and logical development of this study.

For these different types of retaining methods, the most significant concerns are the occurrences of some unexpected accidents during construction work. Accidents are usually caused by an insufficient preliminary design of the retaining system. Liu (2020) proposed some sustainability indicators, such as risk mitigation and reliability, ecology, environmental protection, carbon emission reduction, energy savings, and waste reduction, which are gradually being taken into consideration for greener civil infrastructure development [8]. Some other researchers have also performed studies that took into account the sustainabil-

ity issues related to infrastructure projects [8–16]. This study presents a unique retaining method named the "Anchor Pile with Steel Cable System (APSCS)", which achieved significant sustainable goals. During the foundation excavation of the LPHP, the steel anchors were installed at a 5 to 6 m distance from the excavated surface to serve as the anchor piles for the steel H-shaped retaining columns. To verify that the soil condition was applicable for the APSCS method, the shear-wave velocity test (SWVT) was performed before the LPHP began. The SWVT results showed that the construction site had a high-shear base.

The authors present two other cases in the following section to serve as the comparison studies for the LPHP: the A7 Public Housing Building—Part C (A7PHB-C) and the A7 Public Housing Building—Part D (A7PHB-D). The A7PHB-C and -D are located close to the LPHP and were constructed a little bit earlier than the LPHP. The geological investigation results of the LPHP and the A7PHB-C and -D revealed the similarity of the ground condition.

## 2. Case Studies on the Similar Projects

### 2.1. Case 1: A7 Public Housing Building—Part C (A7PHB-C)

Located in the Linkou district near the LPHP project, the A7 Public Housing Building—Part C (A7PHB-C) is part of another public housing project. The A7PHB included four individual projects (A–D); six buildings each of 18 to 20 stories were built in this project. The contractor of the A7PHB-C project was the same as that for the project that served as the main study of this paper (LPHP), which was New Asia Corp. (NAC), and both of these two projects were contracted on a design–build (DB) basis. The retaining system designed in the A7PHB project used a traditional method, which comprised a vertical retaining column and a horizontal strut H-beam to retain the lateral force of soils during the basement excavation. Three layers of horizontal strut H-beam were used in the A7PHB-C project. The excavation work has a depth of 11.2 m and the total duration of the basement construction was 222 days. Table 1 shows some construction information for A7PHB-C.

**Table 1.** Summary of the excavation-related work for A7PHB-C.

| Excavation Items | Sum/unit | Remarks |
| --- | --- | --- |
| Planned area of excavation | 11,739 m$^2$ | Close to rectangular shape |
| Depth of excavation | 11.2 m | Divided into four layers of excavation |
| Total soil volume of excavation | 134,960 m$^3$ | |
| Steel strut layers | Three layers | Struts with single and double H-shaped struts |
| Total struts weight | 2163 tons | |
| Duration of excavation work | Two months and six days | This included the lean concrete placement and strut erection |
| Basement construction duration | Five months and six days | This included the construction for the first-floor slab and the strut demolition |

The A7PHB-C basement was completed in 2014. During the construction stage of the basement, no accidents or disasters occurred. The retaining method for basement excavation used the traditional horizontal strut system. This method was different from the APSCS method applied in the LPHP. As the H-shapes were not used, the cost of the horizontal steel strut system in the LPHP was reduced by up to NT $350 million. We were aware that the horizontal strut steel members always seriously interfered with the construction work, such as the material hang-in, rebar installation, and concrete placement. This led to a longer construction duration. Thus, as the result of this traditional retaining method, the total construction duration of the basement, including the excavation and structure construction, lasted 7 months and 12 days. This was at least 90 days longer than

the basement construction duration of the LPHP. Figure 3 shows some different stages of the basement construction.

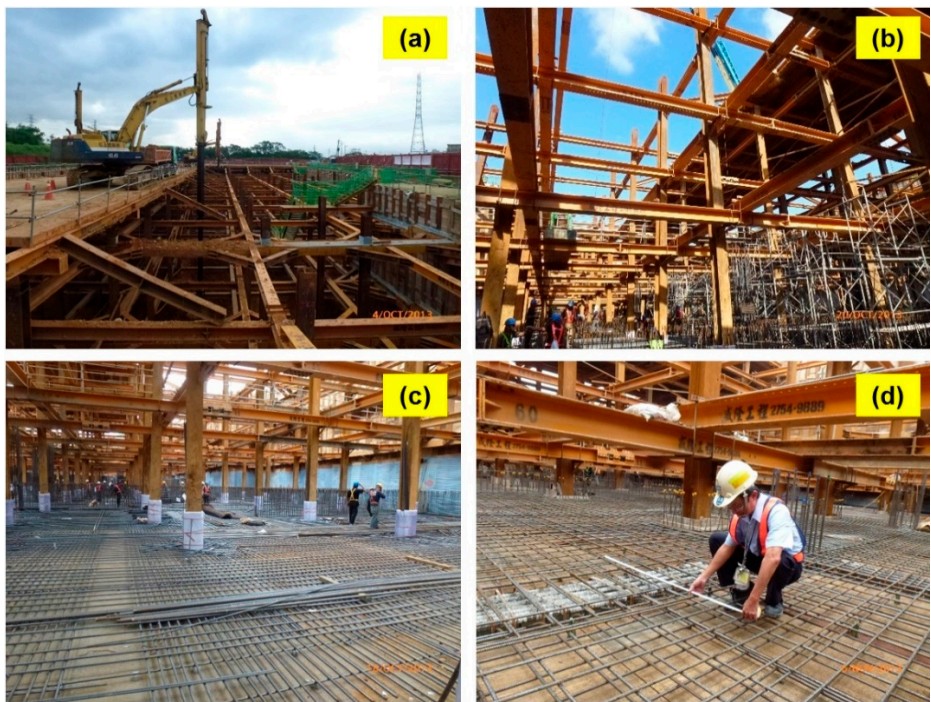

**Figure 3.** Different stages of the LPHP basement construction: (**a**) the soil excavation, (**b**) the horizontal steel strut members and steel pin piles, (**c**) the bottom foundation, and (**d**) the B3 floor inspected by the corresponding author.

From the sustainability viewpoint, longer construction increases indirect costs and raises risk during excavation. Furthermore, the massive quantity of materials used for the retaining system may also increase carbon emissions during basement construction. In this paper, the A7PHB-C case is used as a comparison project for the LPHP.

### 2.2. Case 2: A7 Public Housing Building—Part D (A7PHB-D)

For comparison with the LPHP and A7PHB-C, the authors present the A7PHB-D project, which was the same development project referred to in Case 1. The contractor of this project was different to that of A7PHB-C and LPHP. Basic information in this project, such as the subsurface condition, excavation depth, and groundwater level, was similar to that of A7PHB-C and the LPHP; however, the retaining method selected by the contractor was different in this project from the other two projects. The retaining system was a combination of vertical H-beam columns and soil blocks that were protected by shotcrete. Figure 4 shows the vertical section and the soil profile of the retaining system for the A7PHB-D project.

The major components of this retaining method were the vertical H-beam columns. They were retained by the soil block, which had a height of 9.5 m. There was no horizontal steel strut designed for this retaining system. The total length of vertical steel columns was uncertain. However, site observation showed that the penetration length of the vertical steel columns into the soil was insufficient for the retaining system. Table 2 shows the basement construction information for the A7PHB-D project.

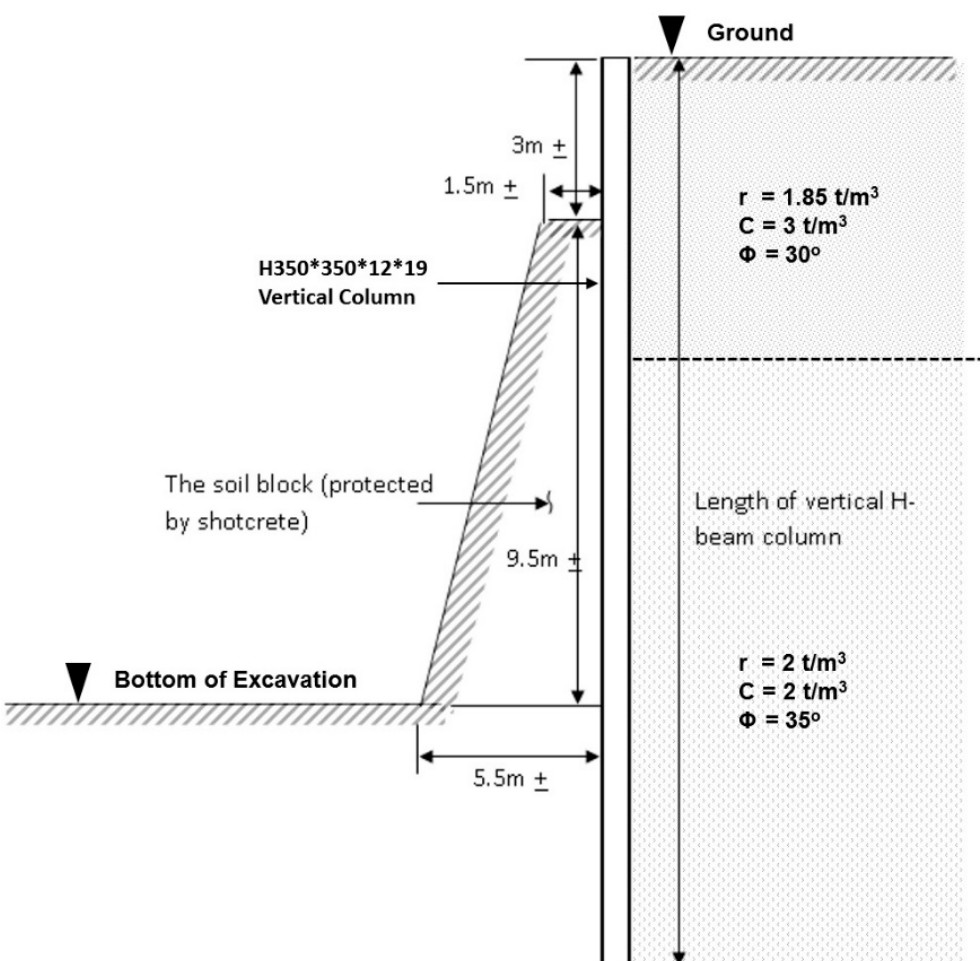

**Figure 4.** The retaining system's vertical section and soil profile for the A7PHB-D project.

**Table 2.** The basement construction information for the A7PHB-D project.

| Excavation Items | Sum/Unit | Remarks |
|---|---|---|
| Depth of excavation | 12.3 m | Divided into four layers of excavation |
| Steel strut layers | No Strut | |
| Duration of excavation work | Seven months and six days | This included the lean concrete placement and reconstruction of the failed retaining steel H-shaped column |
| Basement construction duration | Six months | This included the construction for the first-floor slab |

In this A7PHB-D case, the excavation depth was slightly larger than that of the LPHP. Initially, in order to save on costs, the contractor of A7PHB-D did not implement any horizontal steel struts or a pre-stressed anchor system during the basement excavation case. A free soil block was applied to be the passive earth pressure, as shown in Figure 4. Unfortunately, the failure of the retaining system occurred when excavation work reached the bottom of the basement. Figure 5 shows the severe failure of the steel retaining H-columns in the excavation of A7PHB-D project.

The subsurface contained two layers of soil. The first layer (G.L. 0 to G.L. −6) comprised a reddish-brown silty clay, which was medium to firm, and the second layer (G.L. −6 to G.L. −30) had a very dense laterite gravel combined with the brown silty and sandy clay. Figure 6 shows the distribution of the lateral earth pressure from the soils ($P_{A1}$, $P_{A2}$, $P_{A3}$) and the retaining force exhibited by the soil block (Ws). In this case, the vertical steel columns did not produce many effects in this system.

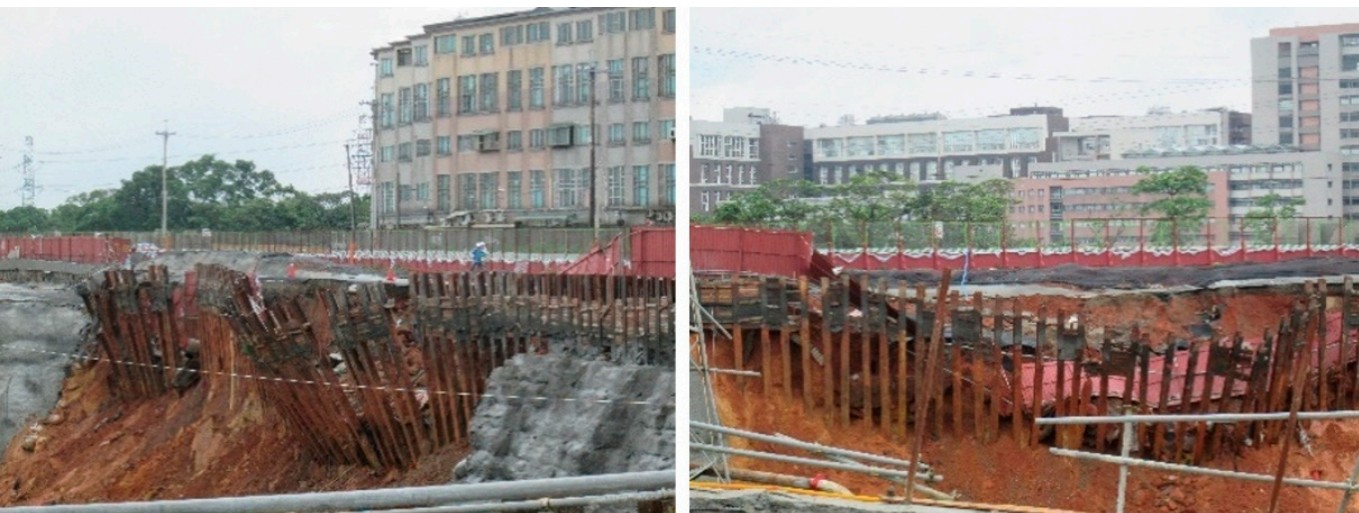

**Figure 5.** Failure of steel retaining H-columns in the A7PHB-D project.

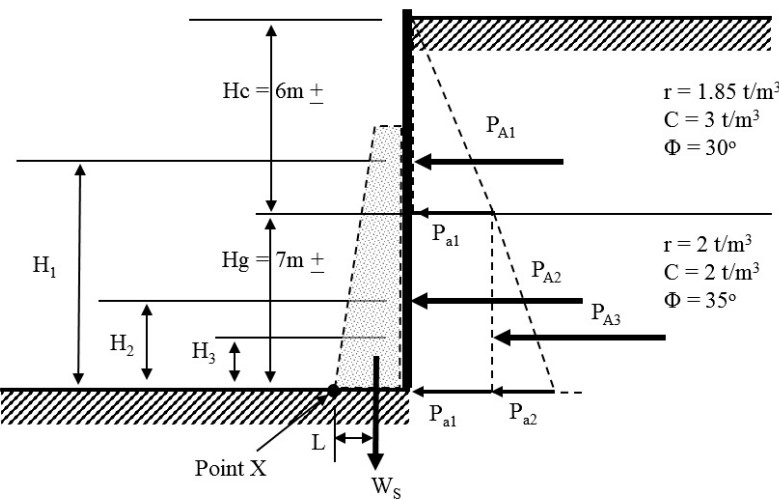

**Figure 6.** The distribution of the soil block's lateral force from the soil and the retaining system.

The equations [17] used to calculate the forces and the computed results are listed in Table 3.

According to the calculated results listed in Table 3, the safety factor obtained for this retaining system was only 0.948 and it was the primary reason for the retaining-system failure. We were aware that the safety factor for the temporary retaining system should have been larger than 1.2 in order to maintain the stability of the soil during the excavation. In this case, insufficient positive earth pressure was applied to the vertical steel H-columns, which could not serve as safe retaining forces for the active earth pressure during the excavation. The absence of horizontal steel struts or an anchor system finally caused the collapse of the steel retaining H-columns at the end of the excavation. This accident resulted in the construction team spending at least five months and ten days reinstalling the retaining system. Fortunately, when the accident occurred, nobody was injured. The knowledge that we gained from this case was that, even in such dense laterit–gravel soil conditions, the failure of retaining system would definitely occur when an improper retaining method was selected for the excavation work.

**Table 3.** The equations used to calculate the forces and the computed results for the A7PHB-D project.

| Parameters or Items | Equations | Value of the First Layer | Value of the Second Layer |
|---|---|---|---|
| Coefficient of at-rest earth pressure, $K_0$ | $K_0 = 1 - \sin\varnothing$ | 0.500 | 0.426 |
| Coefficient of active earth pressure, $K_A$ | $K_A = \tan^2(45^\circ - \varnothing/2)$ | 0.333 | 0.271 |
| Coefficient of passive earth pressure, $K_P$ | $K_P = \tan^2(45^\circ + \varnothing/2)$ | 1.012 | 1.010 |
| Retaining height, Hc/Hg (m) | | 6.000 | 7.000 |
| Dry soil density, $r_d$ (t/m$^3$) | | 1.850 | 2.000 |
| Earth pressure per unit of width due to soil pressure, $P_{a1}$ (t) | $P_a = K_A \times H \times r_d$ | 3.700 | 3.700 |
| Earth pressure per unit of width due to soil pressure, $P_{a2}$ (t) | | | 3.794 |
| Resultant active earth pressure, $P_{A1}$ (t) | $P_{A1} = P_{a1} \times Hc \times (1/2)$ | 11.100 | |
| Resultant active earth pressure, $P_{A2}$ (t) | $P_{A2} = P_{a2} \times Hg$ | | 25.900 |
| Resultant active earth pressure, $P_{A3}$ (t) | $P_{A3} = P_{a3} \times Hg \times (1/2)$ | | 13.279 |
| Weight of the soil block (t) | $V \times rd$ | 70.000 | |
| Gravity center of soil block in relation to the retaining boundary O, L (m) | | 3.000 | |
| Height of pressure $P_{A1}$, $H_1$ (m) | $Hc \times (1/3) + Hg$ | 9.000 | |
| Height of pressure $P_{A2}$, $H_2$ (m) | $Hg \times (1/2)$ | 3.500 | |
| Height of pressure $P_{A3}$, $H_3$ (m) | $Hg \times (1/3)$ | 2.333 | |
| Turning moment produced by $P_{A1}$, $M_1$ (t–m) | $P_{A1} \times H_1$ | 99.900 | |
| Turning moment produced by $P_{A2}$, $M_2$ (t–m) | $P_{A2} \times H_2$ | 90.650 | |
| Turning moment produced by $P_{A3}$, $M_3$ (t–m) | $P_{A3} \times H_3$ | 30.983 | |
| Total turning moment, $M_A$ (t–m) | $M_A = M_1 + M_2 + M_3$ | 221.533 | |
| Turning moment produced by Ws, Ms (t–m) | $Ms = Ws \times L$ | 210.000 | |
| Safety factor | $Ms/M_A$ | 0.948 | |

## 3. Comparison of A7PHB-C and A7PHB-D

For the best realization of the feature differences between the A7PHB-C, A7PHB-D, and LPHP projects, the authors summarized the essential information of these three projects, as shown in Table 4. Please note that a detailed description of the LPHP is provided in the following sections.

**Table 4.** The comparison of essential information between the A7PHB-C, A7PHB-D, and LPHP projects.

| Items | A7PHB-C | A7PHB-D | LPHP | Remarks |
|---|---|---|---|---|
| Soil condition | Laterite–gravel soil (LGS) layer mixed with brown silty and sandy clay | Laterite–gravel soil (LGS) layer mixed with brown silty and sandy clay | Laterite–gravel soil (LGS) layer mixed with brown silty and sandy clay | The soil contents is the same for these three projects |
| Groundwater level | Under excavation bottom level | Under excavation bottom level | Under excavation bottom level | There was no groundwater in the Linkou district |
| Planned area of excavation | 11,739 m$^2$ | 14,295 m$^2$ | 16,060 m$^2$ | |
| Depth of excavation | 11.2 m | 12.3 m | 10.6 m | |
| Steel strut layers | Three layers of struts | No Strut | No Strut | |
| Duration of excavation work | Two months and six days | Seven months and six days | One month and seven days | The duration of A7PHB-D includes the reconstruction of the failed retaining steel H-shaped column |
| Basement construction duration | Five months and six days | Six months | Four months and two days | |
| Retaining-system safety factor | 1.792 | 0.948 | 1.543 | |
| Materials used in the retaining system | 2163 t | 876 t | 342 t | |
| Environmental protection | High carbon emissions cause by the high material consumption | High carbon emissions caused by the collapse accident | Low carbon emissions | |
| The effectiveness of the construction method | Low effectiveness | Low effectiveness | High effectiveness | |

Note: a detailed description for the LPHP is provided in the following sections.

## 4. Description of the Linkou Public Housing Project (LPHP)

The LPHP project was located in the Linkou district, New Taipei City, Taiwan. It was close to the other two cases, A7PHB-C and D, described above. The LPHP included nine buildings; each building contained 19–21 stories. The contract between the client and contractor was a design–build project, and its purpose was to serve as an athletes' village for the 2017 Taipei Summer Universiade. The project boundary's outline shape was somewhat close to triangular, as shown in Figure 7. Table 4 lists some information for the LPHP basement excavation. The construction results listed in Table 5 were based on applying the proposed unique retaining method, APSCS, in this paper.

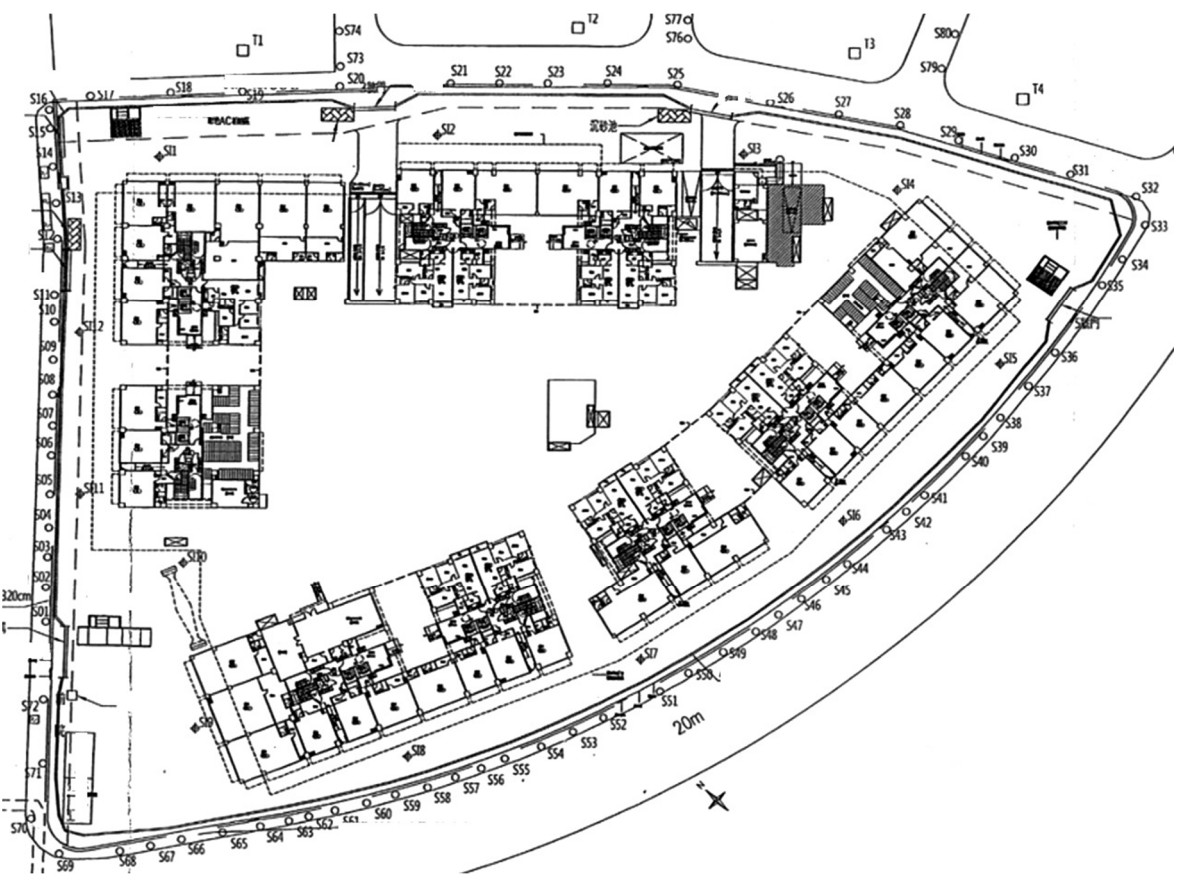

**Figure 7.** Plan view of the LPHP.

**Table 5.** Summary of the excavation-related work for the LPHP.

| Excavation Items | Sum with Unit | Remarks |
|---|---|---|
| Planned area of excavation | 16,060 m² | Close to a triangular shape |
| Depth of excavation | 10.6 m | Divided into four layers of excavation |
| Total soil volume of excavation | 185,500 m³ | |
| Steel strut layers | None strut member | |
| Anchor piles for bearing forces | 97 pcs. | Steel rails of 50 kg grade, L = 13 m |
| Duration of excavation work | One month and seven days | This included the lean concrete placement |
| Basement construction duration | Four months and two days | This included the construction for the first-floor slab |

The soil profile in the Linkou district is a laterite–gravel soil (LGS) layer mixed with brown silty and sandy clay, which appears as a red/brown color, as shown in Figure 8.

The soil combination is mainly LGS, and the groundwater level in this area is 25 m below ground level (i.e., G.L. −25 m) and did not cause any interference with the excavation work. A total of 16 boreholes were made for standard penetration tests (SPTs) and soil samples were gathered for laboratory testing and analysis before the excavation work began [18]. Figure 9 shows the example of the standard penetration test (SPT) results, and Table 6 shows the simplified soil parameters of the LPHP site, which were based on the 16 holes of the SPT results.

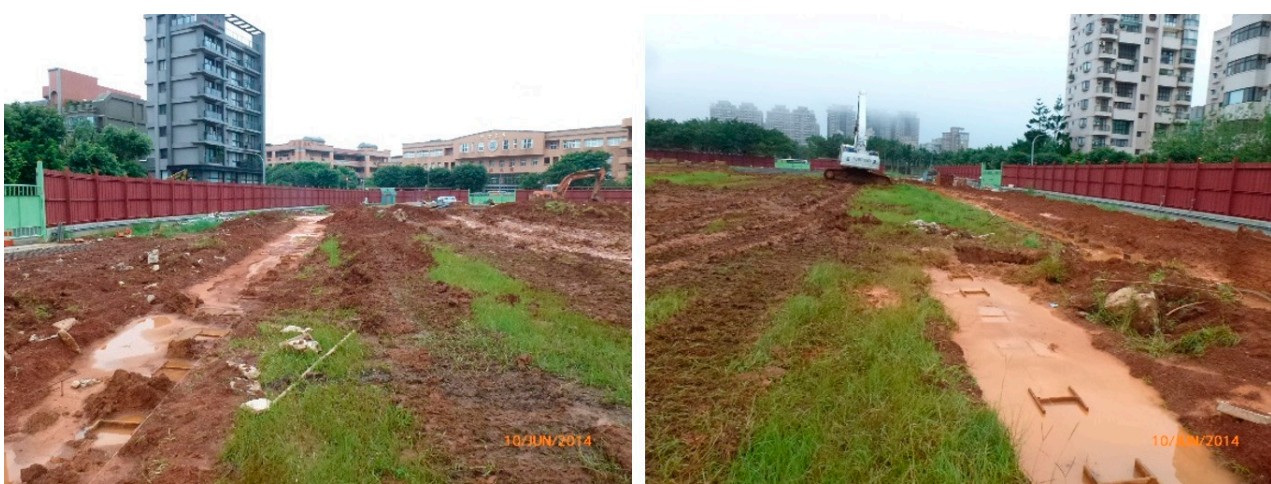

**Figure 8.** The appearances of the laterite–gravel soil (LGS) layer mixed with brown silty and sandy clay in the Linkou district.

**Figure 9.** Example of site SPT results for the LPHP.

**Table 6.** Simplified soil parameters of the LPHP site.

| Layer | Contents | Depth (G.L.) | N Value | Unit Weight | Cc/Cs (T/m$^3$) | Kv (T/m$^3$) | Kh (T/m$^3$) | Su (T/m$^3$) | c (T/m$^3$) | Ø (o) | C' (T/m$^3$) | Ø' (o) |
|-------|----------|--------------|---------|-------------|------------------|--------------|--------------|--------------|-------------|-------|--------------|--------|
| 1 | Clay/ Lateritic–gravel soil | 0~5 | 6~12 (say 8) | 1.84 | 0.2/0.02 | 1600 | 1600 | 4.2 | 2.5 | 15.0 | 0.0 | 29.0 |
| | | 5~9.8 | 12~24 (say 18) | 1.87 | 0.2/0.02 | 3600 | 2230 | 10.8 | 5.5 | 15.0 | 0.0 | 32.0 |
| 2 | Lateritic–gravel soil | 9.8~ | >50 | 2.20 | NP | 8000 | 3380 | NP | 1.0 | 35.0 | 0.0 | 42.0 |

As was already realized, the excavation of the basement construction was typically protected by a vertical retaining system with horizontal steel struts and pin piles, which are shown in Figures 1 and 2. In the LPHP case, due to the triangular shape of the construction site, the horizontal steel strut system was not suitable to be the retaining system. For safety, for the basement excavation of the LPHP the "Anchor Pile with Steel Cable System (APSCS)" method [19,20] was applied; a detailed description of the APSCS is provided in the next section.

## 5. Special Retaining Method for Excavation Work, APSCS

### 5.1. Detailed Description of the Anchor Pile with Steel Cable System (APSCS)

As mentioned in the previous sections, the site's shape led to a challenge in designing and building a horizontal steel strut system. As presented in Section 2.2, a poor retaining-system design could cause failure or disaster during basement excavation. Thus, a special retaining method, APSCS, was designed and established to prevent any unforeseen disasters. Not only was the risk/danger of excavation significantly reduced, higher levels of sustainability were achieved, carbon emissions were reduced, and a circular economy was reached in the LPHP case. Table 7 shows the major components of the APSCS, and Figure 10 shows its cross section.

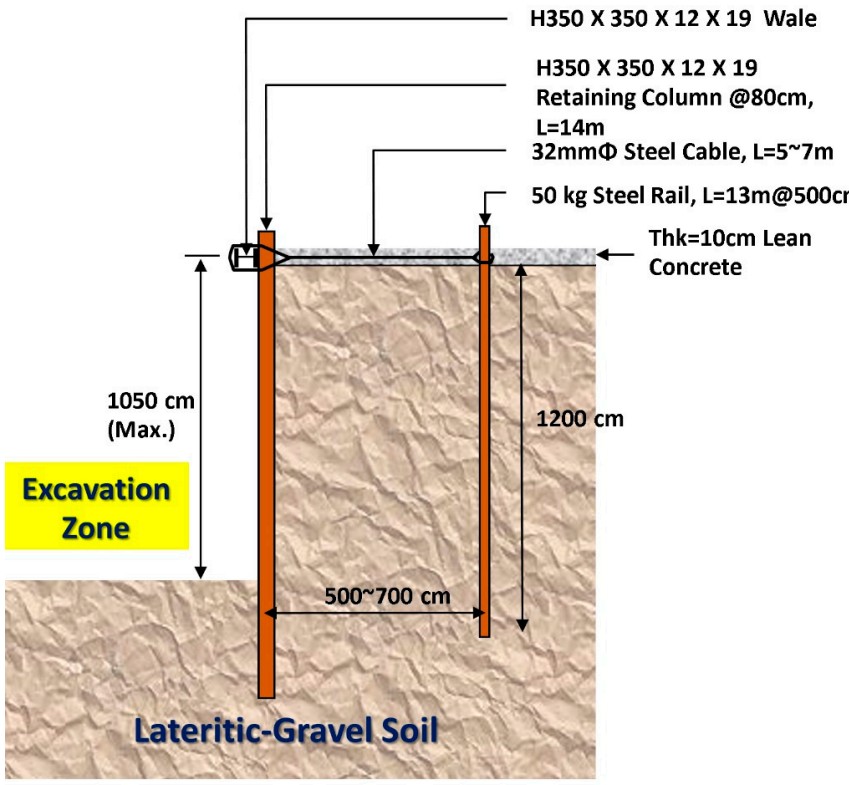

**Figure 10.** Cross section of APSCS method.

**Table 7.** Major components of the APSCS method adopted in the LPHP.

| Items | Description | Quantity | Remarks (G.L. in Reference to Ground Level) |
|---|---|---|---|
| Vertical steel column | H350 × 350 × 12 × 19, L = 16 m | @0.8~1 m | G.L. 0~−16 m |
| Horizontal wales | H350 × 350 × 12 × 19 | 1 | G.L. −0.8 m ± |
| Steel cable | D32 mmΦ, L = 5~7 m | @5~6 m | For forces transferring from the anchor pile to the wall |
| Anchor piles | 50 kg grade steel rail, L = 14 m | @5~6 m | G.L. 0.5~−13.5 m |
| Protective concrete layer | f'c = 140 kg/cm², Thk = 20 cm, W = 5~6 m | On the ground level, around the site | With D = 6 mm wire mesh |

The laterite–gravel soil (LGS) layer might have been softened by contact with water, which would have caused a serious and drastic increase in the active earth pressure and reduced the stability of the retaining system. The risk of collapse of the retaining members would have been increased under this condition. Avoiding the occurrence of this situation, by setting a 20 mm-thick wire-meshed concrete layer to protect against rain water and water from cleaning and other operations, is necessary with an LGS layer, as shown in Figure 11.

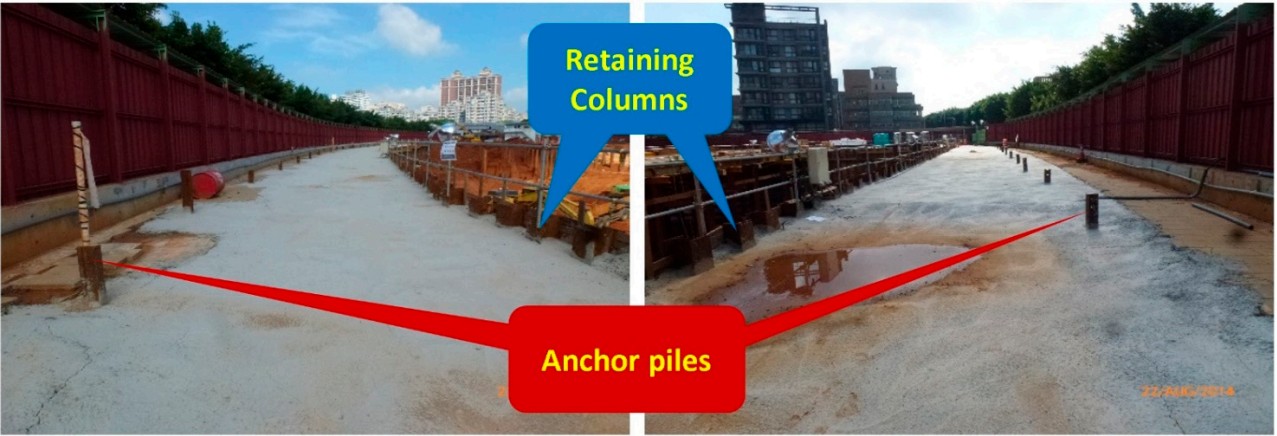

**Figure 11.** The layout of the APSCS members with the wire-mashed concrete protection layer at the ground level.

### 5.2. Analysis by Static Calculation

According to Terzaghi's formula [17], a horizontal soil pressure diagram with a balance condition can be reached, as shown in Figure 12.

With the simplified soil parameters obtained from SPT results in Section 3, the balanced condition of the APSCS cross section is shown in Figure 13 [17].

By adopting the geological equations, the parameters $\gamma_d$, $\psi$, $K_0$, $K_A$, and $K_P$, were then determined, which are listed as shown in Table 8. Table 9 shows the calculated results for the H350 retaining steel columns, including $\sigma$, $\tau$, and $\Delta$.

In Table 9, the equation was adopted to calculate the maximum moment in the H350 column, which was based on the free-body diagram, as shown in Figure 14.

The allowable stress of ASTM A36 materials ($\sigma a$) is 1500 kg/cm², and $\tau a$ is 1000 kg/cm². The $\sigma_{max}$, $\tau_{max}$, and $\Delta_{max}$, as shown in Table 9, confirmed the safety of the retaining-system APSCS method used in the LPHP.

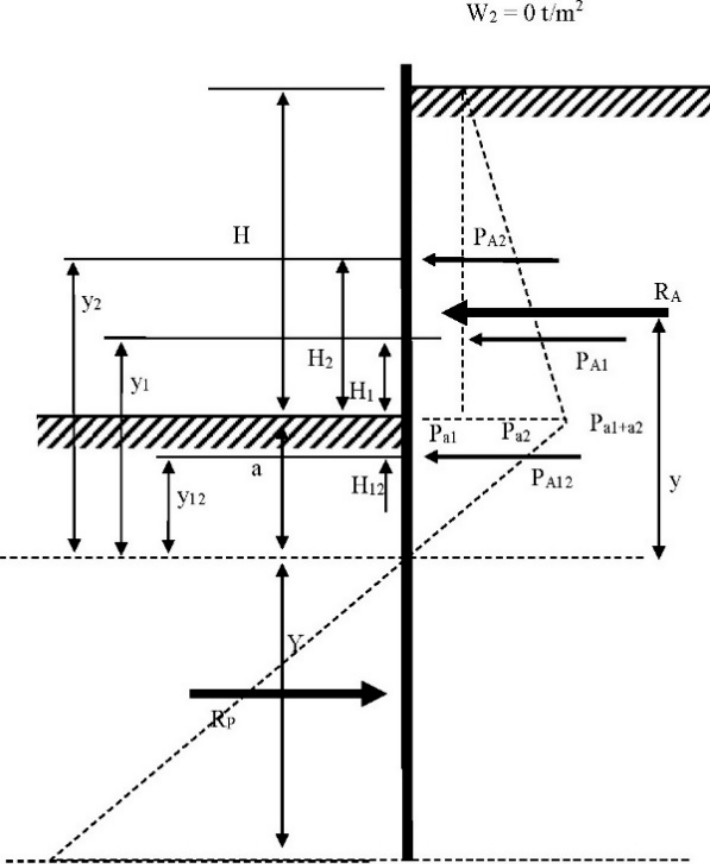

**Figure 12.** Balanced horizontal soil pressure diagram based on Terzaghi's formula.

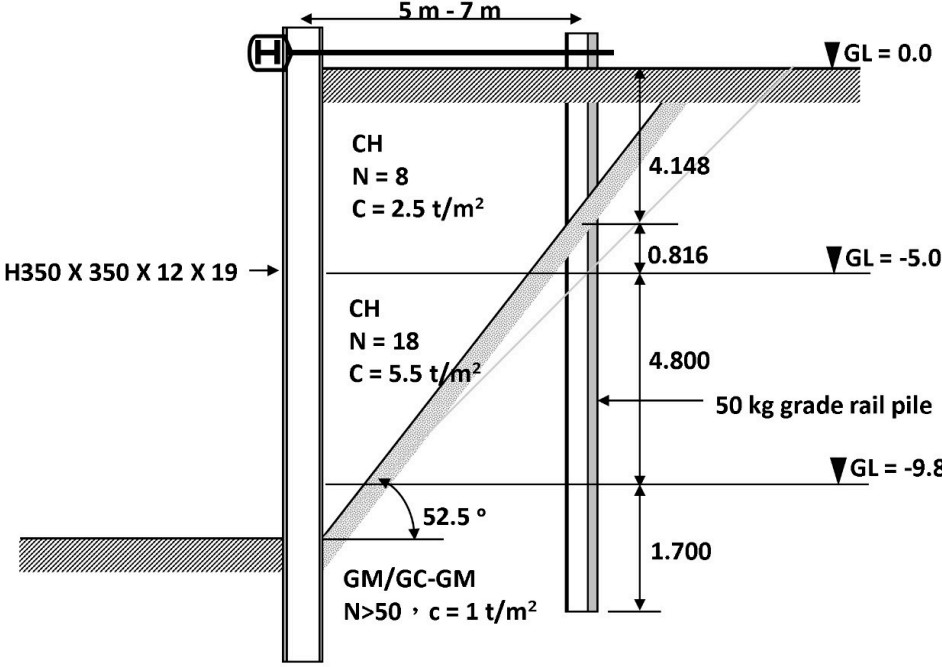

**Figure 13.** The balanced condition of the APSCS cross section.

**Table 8.** The geological equations were adopted to calculate the forces and the computed results for the LPHP project.

| Parameters or Items | Equations | Calculated Value |
|---|:---:|:---:|
| Dry soil density, rd ($t/m^3$) | | 2.0 |
| The angle of internal friction (°) | | 35.0 |
| Surcharge | | 0.0 |
| Retaining height, H (m) | | 9.6 |
| Coefficient of at-rest earth pressure, $K_0$ | $K_0 = 1 - \sin\varnothing$ | 0.4264 |
| Coefficient of active earth pressure, $K_A$ | $K_A = \tan^2(45° - \varnothing/2)$ | 0.271 |
| Coefficient of passive earth pressure, $K_P$ | $K_A = \tan^2(45° + \varnothing/2)$ | 3.69 |
| Earth pressure per unit of width due to soil pressure, $P_{a1}$ (t) | $P_{a1} = K_A \times H \times r_d$ | 5.203 |
| Earth pressure per unit of width due to soil pressure, $P_{a2}$ (t) | $P_{a2} = K_0 \times H \times r_d$ | 0.0 |
| $P_{a1} + P_{a2}$ | | 5.203 |
| Resultant active earth pressure, $P_A$ (t) | $P_A = P_{a1} \times H \times (1/2) + P_{a2} \times H$ | 24.974 |

**Table 9.** Calculated results for the H350 retaining steel columns applied in the APSCS LPHP including $\sigma$, $\tau$, and $\Delta$.

| Parameters or Items | Equations | Calculated Value | Remark |
|---|:---:|:---:|:---:|
| Maximum lateral force, $P_{max}$ (kg) | $P_{max} = P_A \times 1000$ | 24,974 | |
| H350 column length, L (cm) | $L = H \times 100$ | 960 | |
| Height of H350, $H_h$ (cm) | $H_h = 350/10$ | 35 | |
| Cross sectional area of H350, $A_{rea}$ ($cm^2$) | | 173.87 | |
| Second axial moment of H350 shape, Ix ($cm^4$) | $Ix = B_h \times H_h^3/12$ | 40,295 | |
| Maximum moment in H350 column, Mmax (kg-cm) | $P_{max} \times a \times b^2 \times (2 \times L + a)/(2 \times L^3)$ | 2,367,905 | Please refer to Figure 13 |
| Maximum moment stress, $\sigma_{max}$ ($kg/cm^2$) | $\sigma_{max} = M_{max} \times (H_h/2)/Ix$ | 1028 | |
| Maximum shear stress, $\tau_{max}$ ($kg/cm^2$) | $\tau_{max} = P_{max}/A_{rea}$ | 144 | |
| Maximum horizontal deflection, $\Delta_{max}$ (cm) | $\Delta max = P_{max} \times L^3/(48 \times E \times Ix)$ | 5.44 | $E = 2.1 \times 10^6$ |

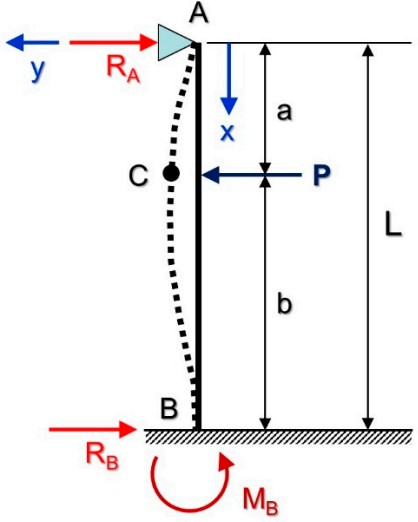

**Figure 14.** The free-body diagram for the calculation of the maximum moment in the H350 column.

### 5.3. Equipped Monitoring Results during and after the Excavation

A total of six inclinometers were installed to monitor the horizontal displacements of the H350 retaining columns during the excavation stage. Figure 15 shows the diachronic graphics of the horizontal displacements measured by the No.3 inclinometer.

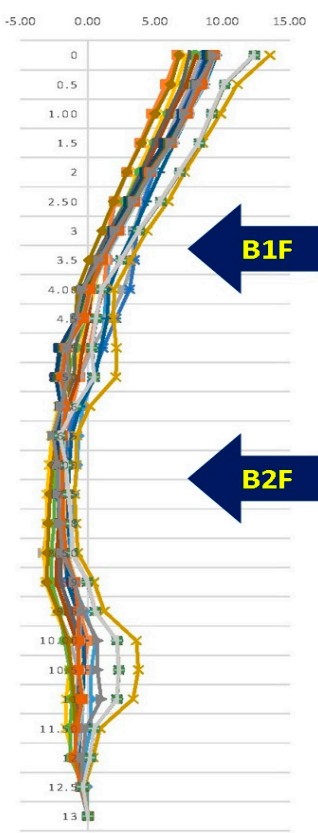

**Figure 15.** Diachronic graphics of the horizontal displacements monitored by the No.3 inclinometer.

It was clear from Figure 14 that the maximum horizontal displacement at the top of the H350 steel columns was 13.71 cm. This value was obviously larger than the $\Delta_{max}$, which was 5.44 cm, as listed in Table 9. This was caused by the horizontal displacement of the top of the anchor pile. The engineers judged that no safety issue needed to be addressed during that stage. The value of this horizontal displacement on the anchor pile top was measured to be 8.27 cm.

### 5.4. Carbon Emission Reduction for Sustainability and the Circular Economy Issue

Compared to Case 1, A7PHB-C, as described in Section 2.1, the APSCS method achieved effective carbon reduction and exhibited the principles of a circular economy because fewer materials were used. Table 10 lists the carbon emission reduction results achieved by adopting the APSCS.

Furthermore, when compared to Case 1, the APSCS method reduced the construction costs by up to NT $350 million and shortened the construction duration by at least 90 days. The APSCS also successfully achieved the principle of the circular economy. Most importantly, without any horizontal struts in the basement construction zone, the APSCS method provided a safer site environment for the installer/worker to perform construction work. Compared to Case 2, the A7 public housing building project, Part D (A7PHB-D), mentioned in Section 2.2, the proposed APSCS method successfully prevented the occurrence of any accidents or disasters during the excavation of the basement. Figure 16 shows the construction site after excavation work (under the corresponding author's supervision)

and the layout of the APSCS members with the wire-meshed concrete protection layer at the ground level, respectively.

**Table 10.** Carbon emission reduction results achieved by adopting the APSCS.

| No | Items | Unit | Summary | Carbon Emission Factor | Carbon Reduction (kg) | Remark |
|----|-------|------|---------|------------------------|-----------------------|--------|
| 1 | Strut | Kg | 2,550,000 | 2.42 | 617,100 | Material ratio = 10% |
| 2 | Transportation | t-km | 99,500 | 0.24 | 23,800 | |
| 3 | Diesel fuel (fixed location) | L | 3100 | 3.42 | 10,602 | |
| 4 | Diesel fuel (moved location) | L | 4320 | 3.45 | 14,904 | |
| 5 | Gas fuel | L | 3500 | 3.10 | 10,850 | |
| 6 | Power | set | 350 | 0.69 | 242 | |
| | Total | | | | 677,578 | |

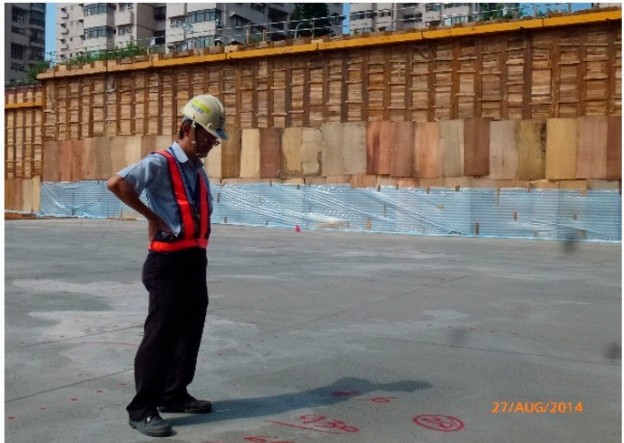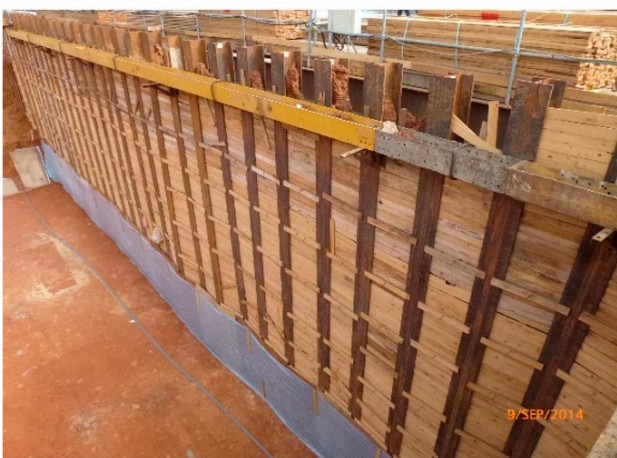

**Figure 16.** The construction site after excavation using the APSCS method (under author's supervision).

## 6. Conclusions

In this LPHP project, the application of the APSCS not only had the risk of disaster been prevented, but also the carbon emissions had been reduced by up to 677,578 tons, and the principle of a circular economy had been achieved due to the decrease in the amount of construction materials used. Under a similar soil condition, groundwater level, and excavation depth, the construction duration of the LPHP was less than those of A7PHB-C and A7PHB-D, which were 119 days and 179 days, respectively. Furthermore, the steel H-shaped materials used for the retaining system of the LPHP were significantly reduced by up to 1800, compared to the traditional retaining method adopted in the A7PHB-C project. The unique retaining system, the APSCS method, executed the excavation of the LPHP foundation with stability and safety. The direct/indirect cost of construction was reduced by up to NT $350 million, and the duration was reduced by up to 90 days. Any possible accidents or disasters were also prevented. The authors concluded that the special APSCS method is a successful, reliable, and functional method to serve as the retaining system for basement excavations.

**Author Contributions:** Conceptualization, T.-Y.L. and S.-J.H.; software, H.-P.T.; validation, H.-K.T.; formal analysis, T.-Y.L. and S.-J.H.; investigation, T.-Y.L., S.-J.H., and H.-P.T.; resources, T.-Y.L.; data curation, T.-Y.L. and S.-J.H.; writing—original draft preparation, T.-Y.L. and S.-J.H.; writing—review and editing, H.-P.T. and H.-K.T.; supervision, H.-P.T. and H.-K.T.; project administration, T.-Y.L. and H.-K.T.; All authors have read and agreed to the published version of the manuscript.

**Funding:** This research received no external funding.

**Institutional Review Board Statement:** Not applicable.

**Informed Consent Statement:** Not applicable.

**Data Availability Statement:** Not applicable.

**Acknowledgments:** The authors would like to express special appreciation to all leaders of the New Asia Construction and Development Corporation. Furthermore, their efforts on disaster prevention, carbon emission reduction, and circular economies have successfully achieved many outstanding results during the past few years. These results were of the utmost important for the completion of this paper.

**Conflicts of Interest:** The authors declare no conflict of interest.

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
