# Peer review of "Using a Unique Retaining Method for Building Foundation Excavation: A Case Study on Sustainable Construction Methods and Circular Economy"

_buildings, doi:10.3390/buildings12030298_

Round 1

Reviewer 1 Report

Through a case study of the Linkou Public Housing Project in New Taipei City, Taiwan, the manuscript reports on the use of a novel retaining method for building foundation excavation. In general, the study has several fascinating datasets that have not been fully utilised. The following shortcomings should be addressed by the authors:

Abstract

Line 29-30: Sentence starting with “The authors…” is incomplete

Add "Case study" to your keyword list.

Introduction

The introduction section lacks a logical flow. It requires substantial rewriting for clarity and grammar.

Special Retaining Method for Excavation Work, APSCS

Include a sub-section for comparing the various retaining types/methods used in the case studies

The grounds for comparison should be explained clearly.

It would be interesting to provide a comparison table that considers aspects such as evacuation items, unit, depth, duration, and so on.

Conclusion

The conclusion section lacks depth. It requires considerable revision to support the findings.

Author Response

This manuscript had been revised in accordance with the reviewer’s comments as follows
Abstract
Line 29-30: Sentence starting with “The authors…” is incomplete
Add "Case study" to your keyword list.
Response: The authors rewrote the incomplete sentence and add the “Case study” to the keyword list.

Introduction
The introduction section lacks a logical flow. It requires substantial rewriting for clarity and grammar.
Response: The authors rewrote the description of the Introduction section and added Figure 2 to illustrate the research framework and; logical development of this study.

Special Retaining Method for Excavation Work, APSCS
Include a sub-section for comparing the various retaining types/methods used in the case studies
The grounds for comparison should be explained clearly.
It would be interesting to provide a comparison table that considers aspects such as evacuation items, unit, depth, duration, and so on.
Response: The authors added to section 3, which contains Table 4 to show the comparisons of the various retaining types/methods used in the case studies.

Conclusion
The conclusion section lacks depth. It requires considerable revision to support the findings.
Response: The authors rewrote the description of the Conclusion section to present the findings in this study.

Reviewer 2 Report

Well-structured manuscript with the representation of case study in a construction site. This manuscript could be interesting reading for civil engineers. 

Author Response

Thank you very much.

Round 2

Reviewer 1 Report

Lines 60-61: Rewrite the sentence beginning "The first author..." My suggestion is to replace the phrase "the first author of this paper, Dr. Liu (2020)" with "Liu (2020) proposed..."

Author Response

Thank you very much. Line 60-61, the sentence begins in "The first author..." had been revised to "Liu (2020) proposed..."
